



# Comparing Airborne Algorithms for Greenhouse Gas Flux Measurements over the Alberta Oil Sands

Broghan M. Erland[1], Cristen Adams[2], Andrea Darlington[3], Mackenzie L. Smith[4], Andrew K. Thorpe[5], Gregory R. Wentworth[2], Steve Conley[4], John Liggio[3], Shao-Meng Li[3], Charles E. Miller[5], John A. Gamon[1,6]

[1]Department of Earth and Atmospheric Sciences; & Department of Biological Sciences, University of Alberta, Edmonton, AB, T6G 2R3, Canada.
[2]Resource Stewardship Division, Alberta Environment and Parks, Edmonton, AB, T5J 5C6, Canada.
[3]Air Quality Research Division, Environment and Climate Change Canada, Toronto, M3H 5T4, Canada.
[4]Scientific Aviation, Inc., Boulder, CO, 80301, USA.
[5]Jet Propulsion Laboratory, California Institute of Technology, Pasadena, California, 91109, USA.
[6]School of Natural Resources, University of Nebraska-Lincoln, Lincoln, NE, 68583, USA.

*Correspondence to*: Broghan M. Erland (erland@ualberta.ca)

**Abstract.** To combat global warming, Canada has committed to reducing greenhouse gases (GHGs) 40-45% below 2005 emission levels by 2025. Monitoring emissions and deriving accurate inventories are essential to reaching these goals. Airborne methods can provide regional and area source measurements with small error if ideal conditions for sampling are met. In this study, two airborne mass-balance box-flight algorithms were compared to assess the extent of their agreement and their performance under various conditions. The Scientific Aviation, SciAv Gaussian algorithm and the Environment and Climate Change Canada Top-down Emission Rate Retrieval Algorithm (TERRA) were applied to data from five samples. Estimates were compared using standard procedures, by systematically testing other method fits, and by investigating the effects on the estimates when method assumptions were not met. Results indicate that in standard scenarios the SciAv and TERRA mass-balance, box-flight methods produce similar estimates that agree (3 - 25%) within algorithm errors (4 - 34%). Implementing a sample-specific surface extrapolation procedure for the SciAv algorithm may improve emission estimation. Algorithms disagreed when non-ideal conditions occurred (i.e., under non-stationary conditions). Overall, the results provide confidence in the box-flight methods, and indicate that emissions estimates are not overly sensitive to the choice of algorithm, but demonstrate that fundamental algorithm assumptions should be assessed for each flight. Using a different method, the Airborne Visible InfraRed Imaging Spectrometer - Next Generation (AVIRIS-NG), independently mapped individual plumes with emissions 5 times larger than the source SciAv sampled three days later. The range in estimates highlights the utility of increased sampling to get a more complete understanding of the temporal variability of emissions and to identify emission sources within facilities.



## 1 Introduction

Global warming is on the pathway to a minimal projected global temperature increase of 3.3 - 5.7 degrees Celsius by 2100 unless meaningful change is enacted to reduce anthropogenic greenhouse gas emissions (Le Quéré et al., 2018; Friedlingstein et al., 2020; Legg, 2021). Anthropogenic carbon dioxide ($CO_2$) and methane ($CH_4$) emissions are the first and second largest
contributors to climate change, respectively (Friedlingstein et al., 2020). Accurate quantification of GHG emissions is an essential foundation for emissions reductions.

Regional, national, and global $CH_4$ and $CO_2$ emissions are estimated using a combination of bottom-up and top-down methods. In general, bottom-up methods aggregate component, or site specific data, and extrapolate to estimate emissions at a larger
scale; whereas top-down methods measure atmospheric GHG concentrations at a larger scale and infer point and area source emissions (National Academies of Sciences, Engineering, 2018). Anthropogenic $CO_2$ and $CH_4$ emissions are estimated using a) ground-based methods, b) airborne methods, or c) satellite methods (e.g., Frankenberg et al. 2016; Conley et al. 2017; National Academies of Sciences, Engineering 2018; Lauvaux 2021; Irakulis-Loitxate et al. 2022). Large differences between bottom-up aggregated inventory estimates and top-down atmospheric budget estimates need to be reconciled to reduce the
uncertainty in estimating global and regional greenhouse gas (GHG) emissions (Nisbet and Weiss, 2010; Dlugokencky et al., 2011; Allen, 2014; Kort et al., 2014; Johnson et al., 2017; Alvarez et al., 2018; Liggio et al., 2019; Saunois et al., 2020; Friedlingstein et al., 2020; MacKay et al., 2021). Improved emission estimates facilitate the best climate change policy, allowing us to adopt pathways for lower global warming increases (Tian et al., 2016; Le Quéré et al., 2018). This paper focuses on airborne approaches as they are intermediate in spatial scale between proximal and satellite sampling methods and therefore
improve estimation by providing essential validation between top-down and bottom-up methods (National Academies of Sciences, Engineering, 2018; Friedlingstein et al., 2020; Cusworth et al., 2020; Nisbet et al., 2020).

While some airborne methods utilize eddy covariance measurements, methods for sampling anthropogenic GHG emissions from the air tend to fall into two major categories: i) mass-balance methods and ii) spectral imaging methods (O'Shea et al.,
2014; Gordon et al., 2015; Yuan et al., 2015; National Academies of Sciences, Engineering, 2018; Wolfe et al., 2018; Duren et al., 2019; Nisbet et al., 2020; France et al., 2021; Tyner and Johnson, 2021; Krautwurst et al., 2021; Foulds et al., 2022). These methods capture atmospheric fluxes using varying approaches that are affected by different biases and are complementary when creating emission budgets. Mass-balance methods quantify the mass flux, or change, in the mixing ratio of a species due to emissions from a known source area. Sampling schemes for mass-balance flights range from flying a single
transect downwind of a source, to multiple stacked transects creating a vertical "screen" to catch the plume at various altitudes, or flying a "box-flight" around a facility, or facility component, to constrain a plume (Gordon et al., 2015; Conley et al., 2017; Baray et al., 2018; Liggio et al., 2019; France et al., 2021). Remote spectral imaging methods fly above potential sources and use absorption spectroscopy, of reflected solar radiance or thermal emissions, to capture regional or facility emissions



(Frankenberg et al., 2016; Bartholomew et al., 2017). Currently, mass-balance box flight methods can attain a lower uncertainty

from a single sample of emission estimates (~ 2 %) than the spectral methods (< 30%), due to smaller background and wind measurement uncertainties (Gordon et al., 2015; Conley et al., 2017; Duren et al., 2019; Thorpe et al., 2020). However, they require understanding of plume sources to know where to fly and they can take longer, therefore they can be more costly.

For mass-balance flights, the location of emission sources must be known. Mass-balance box-flights involve sampling in

stacked, often cylindrical, flight laps, typically surrounding a known source or set of sources, at altitudes varying from the minimum safe flight altitude to the atmospheric boundary layer capping an emission plume (Gordon et al., 2015; Conley et al., 2017). Due to minimum flight height restrictions a gap between the surface and the flight box is inevitable. Concurrent surface sampling is ideal, but often unavailable, so operators aim to fly at a distance from the source where the plume has risen enough, yet has not dispersed to the degree that it cannot be detected, to capture the plume inside the box (Conley et al., 2017).

Extrapolation to the ground is often the largest error source, nearing ~30% when the bottom of the plume is not captured (Gordon et al., 2015; Conley et al., 2017). Mass-balance airborne methods depend on the assumption that the emission plume is captured at the top of the box and does not change during sampling (i.e., that conditions are stationary) (Fathi et al., 2021). Methods applying mass-balance equations to aircraft measurements have been developed and refined over the last two decades (Kalthoff et al., 2002; Alfieri et al., 2010; Karion et al., 2013; Gordon et al., 2015; Conley et al., 2017; Gordon et al., 2018;

Krings et al., 2018; France et al., 2021). Recently, two box-flight mass-balance sampling methods, a Top-down Emission Rate Retrieval Algorithm (TERRA) developed by Environment and Climate Change Canada (ECCC) and a Gaussian theorem algorithm (SciAv) developed by Scientific Aviation, and provide two approaches to evaluate calculate mass fluxes from aircraft measurements (Gordon et al., 2015; Conley et al., 2017). To our knowledge, a detailed comparison of the two methods has not yet been conducted. If algorithm comparisons indicate agreement, then emission estimates from multiple campaigns

using mass-balance and spectral imaging can be aggregated, which will improve the certainty in top-down GHG budgets.

Airborne spectral methods can be considered top-down methods that produce results similar to satellite data, but with higher accuracy (Kort et al., 2014; Frankenberg et al., 2016). Single flights are used as a sample to estimate emissions from sources and repeated sampling can determine source persistence to infer regional emission budgets (Duren et al., 2019). Stationarity

of an emission plume occurs when the source of emission is consistent and meteorological conditions such as the boundary layer and wind are stable throughout the time of sampling. Remote spectral sampling provides quick "snapshots" of features and therefore avoids the stationarity requirement inherent to airborne mass-balance methods, which have lengthy sampling times ranging from less than one hour to multiple hours, depending on the region measured. Remote spectral imaging methods are being advanced by the NASA Jet-Propulsion Lab which has had success in mapping, inferring wind vectors, and estimating

emissions over large areas using their Airborne Visible InfraRed Imaging Spectrometer - Next Generation (AVIRIS-NG)



(Frankenberg et al., 2016; Duren et al., 2019; Jongaramrungruang et al., 2019; Thorpe et al., 2020; Cusworth et al., 2021). Coincident sampling using the SciAv and AVIRIS-NG methods at facilities indicates the methods tend to agree, within errors, when conditions such as the wind speed are similar (Frankenberg et al., 2016; Duren et al., 2019; Thorpe et al., 2020).

Reducing GHG emissions in Canada has become a National and Provincial priority (Johnson and Tyner, 2020; Government of Canada, 2021). Airborne, and ground-based campaigns suggest that the inventories used to facilitate National and Provincial policy are under-reporting GHG emissions (Brandt et al. 2014; Gordon et al. 2015; Johnson et al. 2017; Baray et al. 2018; Liggio et al. 2019; Chan et al. 2020; MacKay et al. 2021; Baray et al. 2021; Tyner and Johnson 2021). Greater certainty in top-down emissions estimates helps flag under-reporting in bottom-up inventories, and better informs GHG policy makers of
emissions, allowing them to enact meaningful GHG reductions. As part of the Joint Oil Sands Monitoring (JOSM) mandate to advance the understanding of Alberta's emissions, a collaborative study was initiated in 2017 by Alberta Environment and Parks (AEP) and the U.S. National Oceanic and Atmospheric Administration (NOAA), contracted to Scientific Aviation, to use instrumented high-performance aircraft to quantify facility-and activity-specific GHG emissions from mineable and *in situ* oil sands developments in northern and east-central Alberta. Between August 2017 and October 2018, sampling was conducted
for various facilities and repeated over several days, to assess both temporal and inter-facility variability in GHG emissions rates.

In this study, we compared emissions estimated using the same data from five box-flights from the 2017-2018 campaign using two airborne, mass balance algorithms (TERRA and SciAv). Our main research objective was to compare emissions estimates
from each algorithm following standard algorithm protocols, and then assess the sensitivity of emissions estimates to surface extrapolation using a variety of schemes. The cause of any differences between the algorithms was assessed. A secondary research objective was to examine the potential for validation of mass-balance emission estimates using complementary AVIRIS-NG spectral imaging data. Our research intention was to assess the comparability of emissions estimates from past campaigns flown by ECCC and SciAv, which may provide greater certainty of GHG emissions from the Alberta Oil Sands
and other regions where these methods are used.





## 2 Methodology

Both mass-balance methods involve flying around a known source in a box pattern to fully capture an emission plume for estimation, but they differ in their approaches (Table 1).


**Table 1. Characteristics of the mass-balance SciAv and TERRA algorithm for box flights[a].**

|  | SciAv | TERRA |
|---|---|---|
| Parameterization of Flux | Simplified to one horizontal flux term. | Quantifies the dynamic system using several flux terms. |
| Conceptual Algorithm Steps | First step: For each lap, solves a single mass-balance integral equation, derived from a Gaussian theorem, using flight measurements decomposed into one single horizontal flux vector, to estimate the divergence due to an emission source within the box.<br><br>Second step: Bin lap divergence estimates by altitude ranges, estimate an average divergence for each bin, then integrate the bins across the total flight height to produce total emission rate estimates. | First step: Applies simple kriging to interpolate flight lap measurements to a spatially resolved screen.<br><br>Second step: Simultaneously solves two mass-balance equations with multiple integrals to fully constrain the system to evaluate a total emission rate estimate. The first equation quantifies emission flux using seven integral terms and the second has three air flux integrals to account for air flow. |
| Surface Extrapolation | Extends lowest bin average divergence as a constant to the surface. | Spatially resolved screen of mixing ratios extended to the surface using one of five extrapolation options depending on plume character. |
| Output | Emission rate from point or area source. | Emission rate from point or area source. |
| Fundamental Assumptions | Stationary plume, stable meteorological conditions, and full plume capture at the top of the plume. | Stationary plume, stable meteorological conditions, and full plume capture at the top of the plume. |
| Error Terms | Three broad terms. | Seven specific terms. |

[a] For the theory, equations, and derivations, see Conley et al. 2017 for the SciAv and Gordon et al. 2015 for the TERRA algorithms, respectively.




## 2.1 Box-Flight Aircraft Measurements

The AEP-NOAA-Scientific Aviation 2017-2018 Alberta Oil Sands Flight Campaign conducted 150 flight segments at 16 different facilities across Alberta. Many of these facilities include multiple components, such as a plant, a mine, and/or tailings ponds. The aircrafts flew in laps around either the entire perimeter of a facility, or around specific components. The data were

collected and processed by Scientific Aviation (Boulder, CO, USA) on contract to NOAA. Flights were performed using two fixed wing, single-engine aircraft, a Mooney M20R (Aircraft N617DH) and a Mooney M20M (Aircraft N2132X) equipped with monitoring equipment. Concentrations of $CO_2$ and $CH_4$ were measured using a cavity ring down spectrometer (Picarro 2401m or 2210m, Picarro Inc., Sunnyvale, CA, USA) in its precision mode at ~0.5 Hz as described by Crosson (2008). Other variables used in the analysis were measured using the airplane primary flight information system and GPS, including wind

speed components (m s$^{-1}$), pressure (mb), temperature (K), heading (deg), altitude (m), and latitude and longitude. Each flight segment was screened to assess whether: (i) sufficient altitude was reached to capture the entire plume, (ii) winds were sufficiently strong and consistent, and (iii) upwind sources were negligible relative to emissions inside the box. If these criteria were not met, then the algorithm results (i.e., emissions estimates) were considered unreliable.

Five flights from three facilities were selected for the algorithm comparison, and are summarized in Table 2, with sample codes (F01 to F05) assigned for comparison purposes. The three facilities from the Athabasca Oil Sands Region included in the study were: Mildred Lake and Aurora North Plant Sites (Syncrude), Horizon Oil Sands Processing Plant and Mine (CNRL), and Suncor Energy Inc. Oil Sands (Suncor). Flight paths around the facilities are shown in Figure 1. These five flights were chosen to capture a range of possible sample types given varying profile shapes, number of laps, boundary layer height, season,

and whether the flight was around a facility perimeter or plant. Four of the five flights selected were considered ideal samples during preliminary flight screening. One flight, F05 was chosen as a poor-quality sample, rejected during flight screening as having 'not fully captured' the top of the emission plume, and was used to assess how the methods compare when a fundamental assumption of the method are not met. Boundary layer height was estimated by Scientific Aviation by assessing profile changes in potential temperature gradients before and after flights. Through the flight screening process, all five flights

were judged to have consistent, stationary winds and stable boundary layers. F02 had normal operating conditions and no flaring events reported by CNRL Horizon. Facilities were informed before sampling. Operating conditions at the oil sands facilities for F01, F03, F04, and F05 were not shared at the time of writing.





**Table 2. Information on the five flight samples used in the comparative analysis.**

|  | F01 | F02 | F03 | F04 | F05 |
|---|---|---|---|---|---|
| Facility Code | Syncrude | CNRL | Suncor | Syncrude | Suncor |
| Area Sampled | Perimeter | Plant | Plant | Plant | Perimeter |
| Date | 2018-04-24 | 2018-07-19 | 2018-04-19 | 2017-08-14 | 2018-09-06 |
| Season | Spring | Summer | Spring | Summer | Fall |
| Min. Altitude (m) | 168 | 173 | 139 | 150 | 157 |
| Max. Altitude (m) | 1057 | 1246 | 775 | 1043 | 563 |
| Boundary Layer Height (m) | 1100 +/- 150 | 900 +/- 200 | 600 +/- 100 | 900 +/- 50 | 500 +/- 100 |
| # Laps | 8 | 14 | 19 | 25 | 7 |
| Start Time (GMT) | 20:17:06 | 20:48:38 | 17:14:36 | 19:11:37 | 17:36:56 |
| End Time (GMT) | 22:47:42 | 21:43:31 | 18:12:12 | 20:09:09 | 19:55:00 |






**Figure 1: The flight path (one second intervals) are shown for each facility sample used in the study. Perimeter flights are the large polygons, and plant flights the smaller ovals. Map layer data © Google Satellite Hybrid 2017.**


Figure 2 shows an example of a close view of a flight path for F04 CH$_4$. Blue dots indicate background levels in ppm and enhanced mixing ratios within the plume are in a gradient of cyan-yellow-red, with the largest enhancements in red. A large plume can be seen on the North-East section of the flight path in Figure 2 and a few enhancements were measured elsewhere

along the flight path. There is evidence that the top of the plume is captured, as dots at the highest altitude show background concentrations, and the flight path goes above the estimated boundary layer.

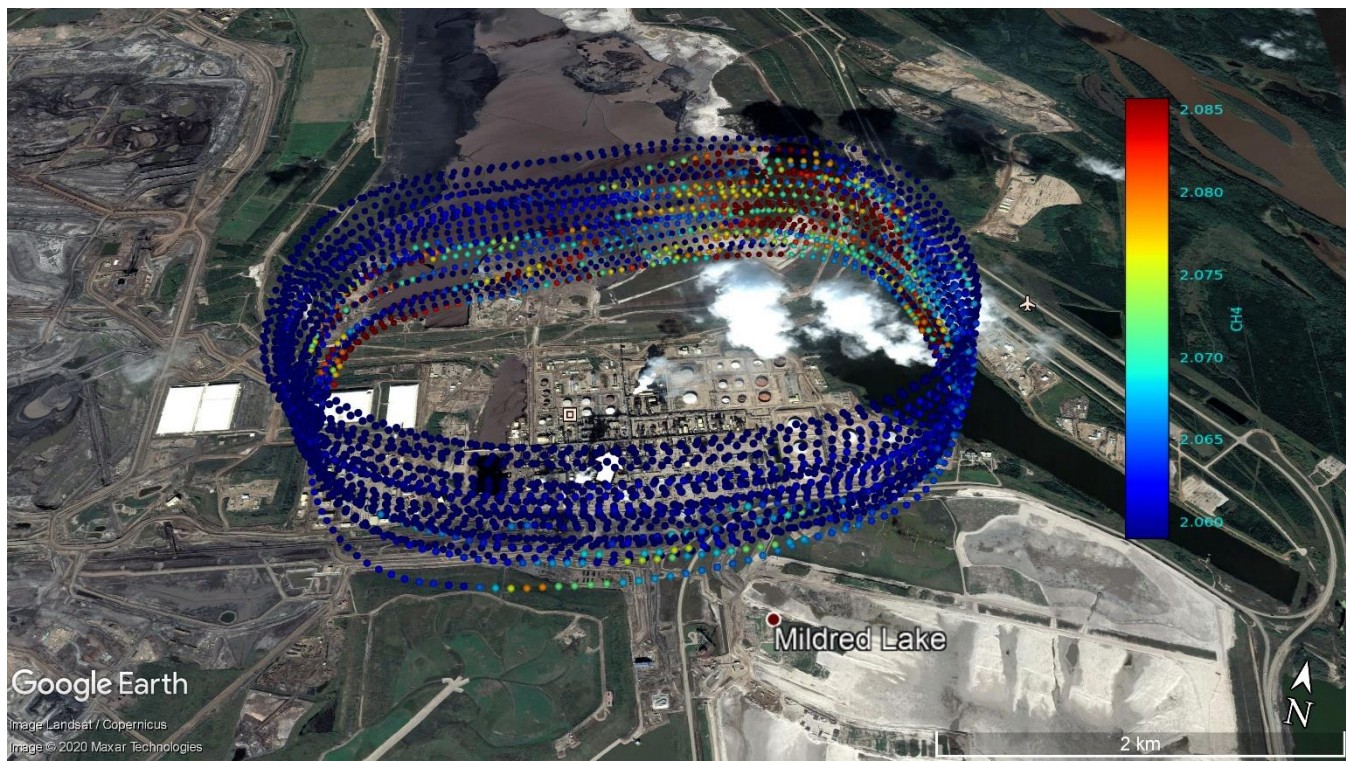

**Figure 2: The flight path for F04 depicts the mixing ratio of CH4 (ppm) measured at 1 Hz intervals for each of the 25 laps around the Syncrude plant. A GoogleEarth historical image from 9/2016 was used as it shows an emission plume with wind conditions similar to the 2017, F04 flight. A KML file containing mixing ratios provided by Scientific Aviation was overlaid on the image. Each measurement of a mixing ratio is depicted as a dot and the layout traces the 25-lap flight path for sampling during F04. Satellite imagery © 2020 Maxar Technologies, Google Earth.**

## 2.2 Box-Flight Emissions Estimate Algorithms

### 2.2.1 TERRA Algorithm

Environment and Climate Change Canada (ECCC) provided the TERRA algorithm, ran portions of the analysis, and detailed instructions on how to produce estimates using the algorithm with commercial plotting software (IGOR Pro 8, Wavemetrics, Lake Oswego, OR, USA). The first step of the TERRA algorithm creates the screen of spatially interpolated lap data by applying simple kriging to the campaign data collected by Scientific Aviation. The spatially resolved screen of mixing ratios is a 2-dimensional unraveling of the lap data by altitude over the length of sampling and is commonly referred to as the 'box' (Gordon et al., 2015).

For the second step, TERRA has 5 options for extrapolating emission concentrations from the lowest flight layer to the surface to account for fluxes below the flight path: 1) A background extrapolation fills all data below the flight path with a background



concentration and (used when there is a fully captured, elevated plume and there is a reason for choosing one single mixing ratio extrapolated to the surface); 2) A constant extrapolation uses the concentration at the bottom of the screen and assumes this remains constant to the surface (best used in the general case of a fully captured, elevated plume as it avoids the assumption of a background value); 3) The linear method fits a line through the lowest points on the screen up to an altitude of 300 m above ground level (preferred in the scenarios when emissions occur from the surface such as a low plume that was not fully

captured, or a mixed plume with ground sources such as a tailings pond); 4) The interpolation between the concentration at the lowest altitude of the screen and the background concentration at the surface (ideally used when there is evidence of decreasing emissions with only a trace of the plume at the bottom of the flight path), and; 5) Exponential extrapolation calculates a Gaussian fit through the lowest points on the screen (largely avoided unless there is a strong argument that it best fits the plume behaviour).


Surface extrapolation was essential for this study as all flights had emission plumes that were not fully captured at the lowest flight track. For the TERRA standard estimates, a linear fit was used for F01, F03, and F05 for both $CH_4$ and $CO_2$ due to their low position on the screen and likelihood of having an increasing emission towards the surface. An interpolation to background fit (i.e. Option 4 described in the paragraph above) was applied to F02 for both $CH_4$ and $CO_2$, and F04 for $CO_2$ as these cases

largely captured plumes with low mixing ratios at the bottom of the flight path. A constant extrapolation (Option 2) was fit to F04 for $CH_4$ to avoid an assumption about the background concentration as it was the one flight with a very large plume dispersion where plume behaviour was unknown (S.I., Section 1.1). All extrapolation outcomes were produced to calculate the surface extrapolation error and to compare with the range of possible outcomes from the SciAv method. The settings for the standard TERRA emissions estimate were chosen by assessing the plume location, boundary layer conditions, and plume

source information to determine the appropriate surface extrapolation (Gordon et al., 2015; Baray et al., 2018).

The TERRA total uncertainty estimate was calculated by adding seven error terms in quadrature (i.e., by taking the square root of the sum of squares). Four of the seven TERRA error terms were evaluated. The wind and measurement error have been previously determined to each be <1%, and the vertical turbulence term has been functionally removed from TERRA analysis

(Gordon et al., 2015; Baray et al., 2018). The surface extrapolation error was calculated as the maximum percent change amongst the plausible surface extrapolation estimates. For example, background extrapolations (Option 1), which assume no mixing ratio enhancement below the flight path, were not considered for standard estimates when a flight has increasing emissions at the bottom of the screen. A description of the calculation of the box-top mixing ratio, air density, and box-top height error terms is given in the Supplementary Information, Section 1.2.

**2.2.1 SciAv Algorithm**

Scientific Aviation provided results from the first step of estimating the divergence for each lap. They applied their algorithm to the flight data and provided output that could be used to address the research objectives. This output included standard



emissions estimates and uncertainties using the SciAv preferred settings. It also included profiles of divergence and uncertainty for each lap versus altitude and preferred bin altitude ranges. These were used in the second step analysis of binning lap

estimates and integration of the flight profiles to test cases such as extrapolation to the surface in MATLAB 2020a. The method for applying the second step of the SciAv algorithm was redeveloped and coded in MATLAB 2020a (The MathWorks, Inc., Natick, Massachusetts, United States).

Figure 3 provides an example profile of the average divergence per lap estimate calculated in the first step of the SciAv

algorithm. A flight is classified as 'fully-capturing' the emission plume when the mean divergence of the highest flight laps approaches zero. The standard SciAv emissions estimates assume that the divergence profile is constant below the lower flight altitude.

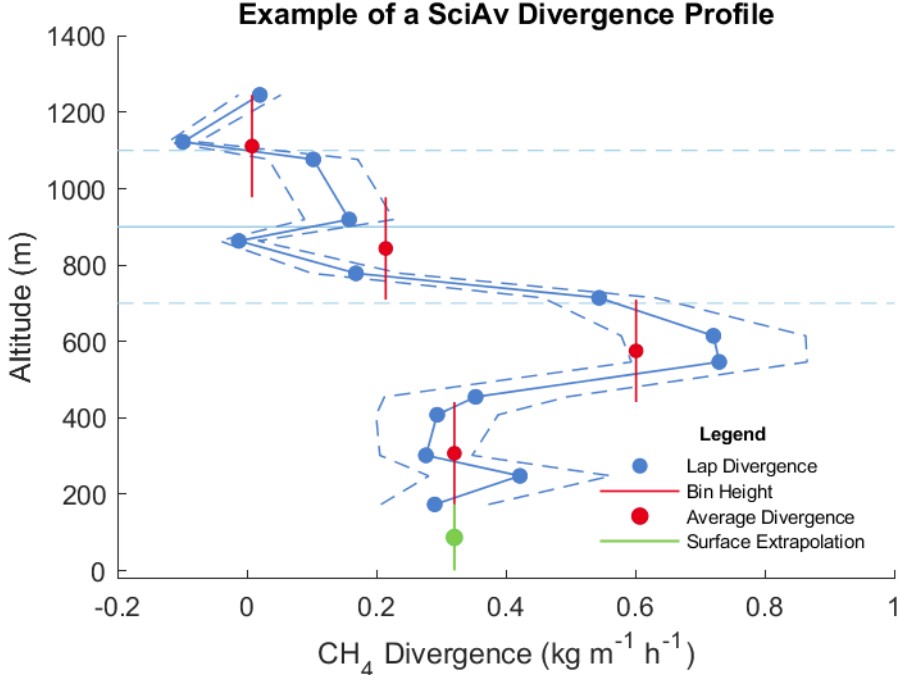

**Figure 3: An example of a SciAv profile used in the second step of the method. The blue points are the estimated divergence for each**
**lap which are connected to show profile shape with the associated error (a dashed blue line). Red points are bin averages and the vertical red bar is the bin height range. The boundary layer height is drawn in light blue with error bars (light blue dashed lines). The standard SciAv surface extrapolation method of extending the lowest red bin to the surface is shown in green.**

For this study, the use of different surface extrapolations for SciAv were developed and tested. To obtain a greater range of

possible outputs from the algorithm, SciAv was fit using differing surface extrapolation methods: 1) constant (SciAv's standard of extending the lowest bin to the surface); 2) background (estimating divergence as zero by applying no extrapolation below the lowest profile point); 3) linear (estimating the linear trend of the profile points at a specific height that was chosen given





the profile shape, location of the plume enhancement, and sparsity of points, and extrapolating the trend to the surface);  4) linear weighted (estimating the linear trend of the profile points, but weighted by the calculated divergence uncertainty for

each lap estimate); 5) linear interpolation to background (fitting a line between the lowest profile bin and , and 6) the average of the surface points calculated in methods 1-5; (S.I., Section 1.3).

The data used for both the TERRA and SciAv methods were identical, but due to different approaches to assessing conditions and analysis of the data, different error estimates were produced. Conley et al. (2017) found that binning by lap and using a

constant extrapolation produces a stable estimate when 20-25 laps are flown around an emission source (Conley et al., 2017). However, with larger area samples, such as perimeter flights, fewer than 10 laps are often flown. These types of samples may be better suited to a different type of integration as well as surface extrapolation. A potential method of improving estimation in the SciAv methods was investigated by using trapezoidal integration rather than binning to estimate a total emission estimate from the lap divergence points.  Figure 4 depicts the two methods of integrating the SciAv divergence profiles, and the different

types of SciAv surface extrapolations, using the profile for F01 CH₄ as an example. The same surface extrapolation estimation procedure was used for both the binning and trapezoidal methods (S.I., Section 1.5). Surface extrapolation methods were fit to the lowest divergence lap point for the trapezoidal method to remain consistent with the SciAv method.

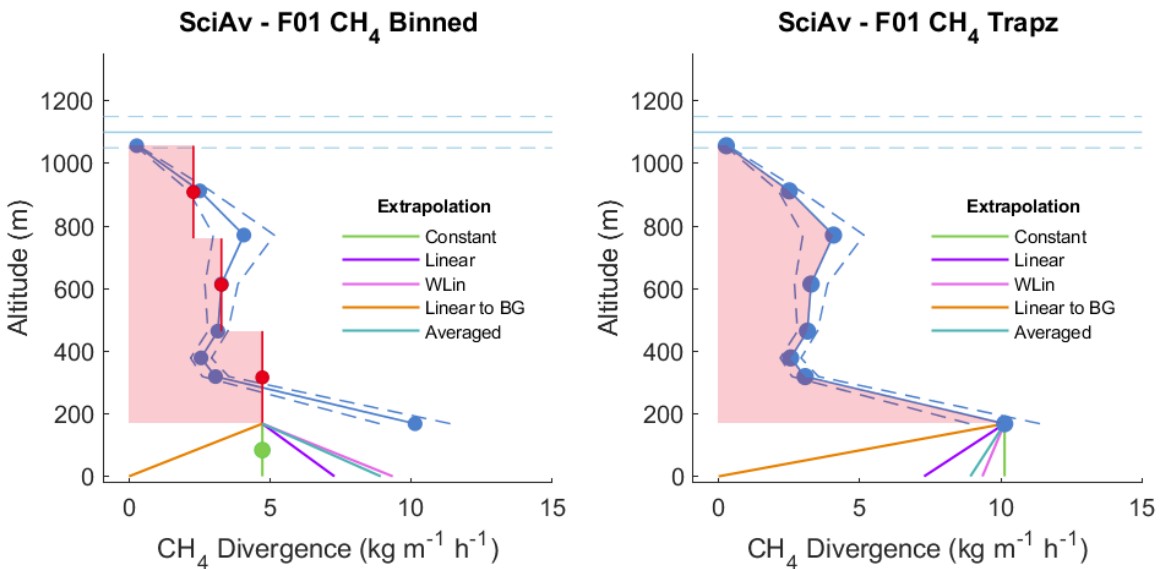

**Figure 4: The two different integration methods applied to the SciAv F01 CH4 profile is depicted as the red area. The left figure shows the standard binning method of estimating an average divergence for each bin and integrating by altitude over the area as rectangular boxes. The right figure shows the trapezoidal method of estimating an average area under the curve by connecting the divergence lap points. The boundary layer height is drawn in light blue with error bars (light blue dashed lines). Both figures do not include the extra emissions that would be included using surface extrapolation.**




## 2.3 AVIRIS-NG Aircraft Emissions Estimates

Three days prior to the F04 flight, a NASA – Jet Propulsion Laboratory (JPL) AVIRIS-NG flight covered the Syncrude plant. This measurement was part of a larger Arctic-Boreal Vulnerability Experiment (ABoVE) which included flight lines flown over the Alberta Oil Sands region. AVIRIS-NG measures ground-reflected solar radiation (380 – 2500 nm) with a 34° field of view, and a spectral resolution of 5 nm to map $CH_4$ plumes by utilizing absorption features in the shortwave infrared (Thorpe et al., 2017, 2020). As described in Duren et al. (2019), emission estimates are calculated by combining the integrated mass enhancement (IME) and wind speed as demonstrated in a number of recent studies throughout the United States (Cusworth et al., 2021, 2022) as well as a controlled release experiment (Thorpe et al., 2021). Figure 5 shows the AVIRIS-NG plume imagery that was captured over the Syncrude plant site with the Scientific Aviation KML lap data (shown in Figure 2) overlaid. By measuring emissions at the same facility within a few days, this independent sample using a different method provides a contrast to the box-flight, mass-balance data.

AVIRIS-NG data were collected at 17.5 kft and the Syncrude facility was not informed prior to sampling. NASA-JPL provided $CH_4$ emissions calculated using AVIRIS-NG data and three sources for hourly estimation of the wind: ECCC meteorological towers 3062696 and 3062697, and MERRA2 (Modern-Era Retrospective analysis for Research and Applications, version 2) reanalysis. MERRA2 is an atmospheric reanalysis method produced by NASA that utilizes numerous satellite observations to produce a global time series of atmospheric data (Gelaro et al., 2017). To estimate variability in wind speed, an average over a three-hour window was used for the met tower data, and nine kernels centred on the plume latitude and longitude were used for the MERRA2 analysis. The magnitude of the AVIRIS-NG estimates was then compared to the SciAv estimate.



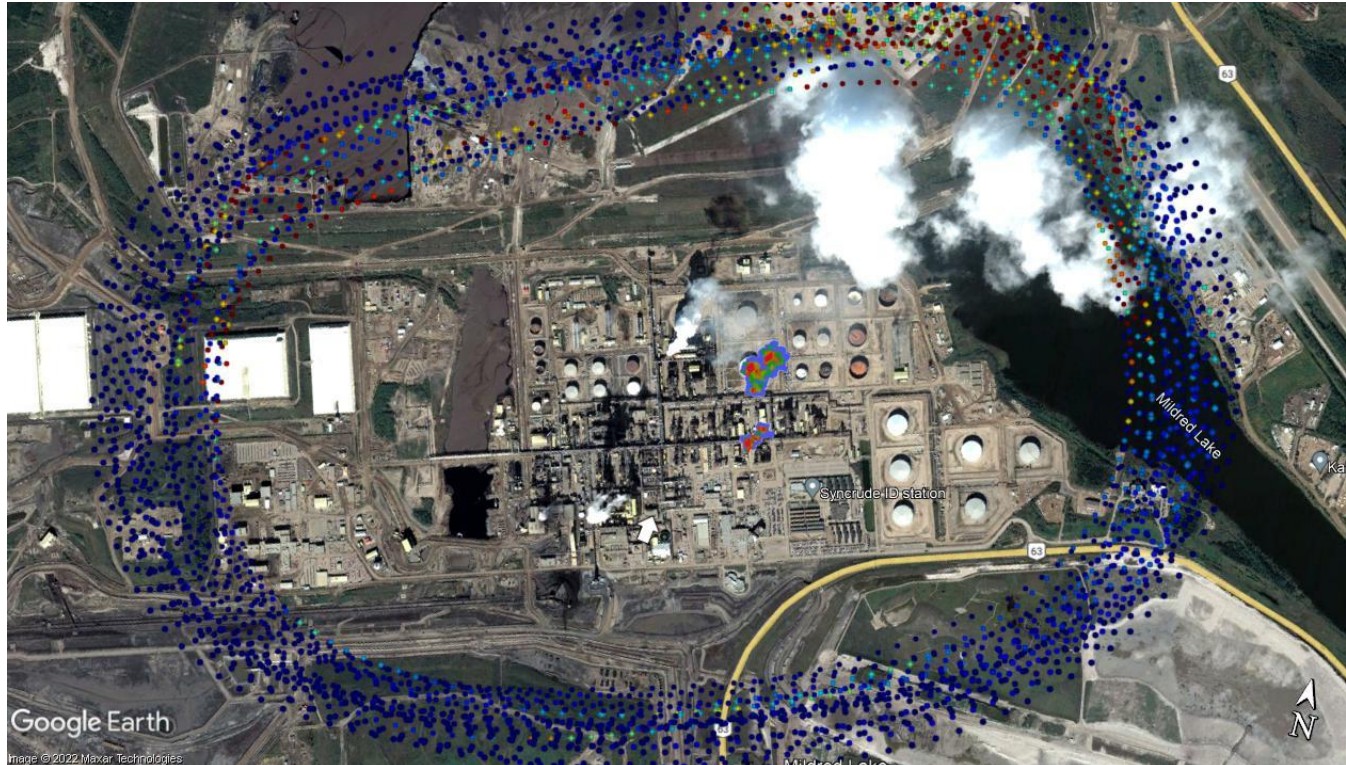

**Figure 5: AVIRIS-NG captured CH₄ column enhancements are shown from 2017-08-11 inside the F04 raw CH₄ lap data around the Syncrude plant from SciAv for 2017-08-14 using the GoogleEarth historical image from 9/2016. AVIRIS-NG data and imagery provided by the NASA-JPL and satellite imagery © 2020 Maxar Technologies, Google Earth. Large CH₄ enhancements are depicted in red. The wind direction is shown by the white arrow as measured by SciAv.**

## 3 Results

### 3.1 Box-Flight Emissions Estimate Comparisons

Standard Scientific Aviation (SciAv) emission results were compared to the estimates produced by applying TERRA to the same flight data. A constant extrapolation to the surface was used for all SciAv samples, whereas the extrapolation for TERRA varies by the flight profile and source. The standard estimate results from both algorithms are shown in Table 3 and Figure 6. Standard emission estimates for four of the five flights agree within their errors. Confidence intervals for the estimates were not produced as there is only one estimate for each flight, and therefore the error bars are simply the range for each estimate. In Figure 6, the error bars for each estimate overlap with each other, aside from F04 which has a large gap between estimates. The errors for the TERRA estimates are consistently smaller than for SciAv (averaging ~ 8 % smaller). Algorithm agreement is implied when the range for each estimate overlap. For all flights except F04, the differences between the algorithms are in the range of the estimate errors. For F04 the emissions estimates disagreed as there is a large gap between the estimates with no overlap of the ranges. To compare the estimates using the error range, the relative mean percentage difference and





propagated percentage error of the two estimates were calculated. The whole set of results from five flights were formally tested for differences between the SciAv and TERRA estimates using a weighted t-test and Wilcoxon signed rank test. As a collective, the differences between the algorithms were found to be insignificant for both $CH_4$ and $CO_2$ (S.I., Section 1.4).

**Table 3. Results of the $CH_4$ and $CO_2$ standard fit estimate from each algorithm with their error as a percentage ± % and the range derived from the percentage error of each standard estimate.**

| Estimate Type | F01 | F02 | F03 | F04 | F05 |
|---|---|---|---|---|---|
| SciAv $CH_4$ (kg h$^{-1}$) | 3840 ± 18% | 362 ± 21% | 497 ± 17% | 349 ± 11% | 3470 ± 29% |
| Range (kg h$^{-1}$) | (3150 - 4540) | (287 - 437) | (415 - 579) | (310 - 387) | (2480 - 4470) |
| TERRA $CH_4$ (kg h$^{-1}$) | 4810 ± 11% | 395 ± 6% | 476 ± 5% | 125 ± 18% | 3910 ± 15% |
| Range (kg h$^{-1}$) | (4310 - 5300) | (373 - 418) | (452 - 501) | (102 - 148) | (3330 - 4490) |
| SciAv $CO_2$ (t h$^{-1}$) | 1040 ± 22% | 563 ± 18% | 526 ± 11% | 1170 ± 11% | 850 ± 34% |
| Range (t h$^{-1}$) | (807 - 1270) | (464 - 662) | (469 - 583) | (1040 -1300) | (561 - 1140) |
| TERRA $CO_2$ (t h$^{-1}$) | 1340 ± 10% | 515 ± 6% | 467 ± 4% | 569 ± 7% | 877 ± 26% |
| Range (t h$^{-1}$) | (1200 - 1470) | (486 - 545) | (451 - 483) | (561 - 60) | (650 - 1110) |




**Figure 6: Flight emission estimates for CH$_4$ (left) and CO$_2$ (right) derived for each algorithm are plotted as points, along with the range of each estimate, as error bars. TERRA standard estimates are shown in green and SciAv in purple. Samples of F01, F02, F03, and F05 produce estimates that coincide within each method's error bars.**

Large, anomalous differences between the SciAv and TERRA estimates occurred for F04. During screening of the flights, no issues with F04 were flagged (S.I. Section 1.6). This flight can be used as an example for improving SciAv flight screening and for assessing the implications when assumptions of a stationary plume and stable meteorological conditions are violated. No other flights had non-stationary conditions. The flight that was intentionally included as a poor-quality sample (F05), due to the large emission plume occurring at the highest altitude transects flown, has very good agreement between the two algorithms.



During the study design, F04 was considered an ideal sample, and the non-stationarity of the emission plume was not flagged
until more in-depth analysis was applied to discern the reason for the large disagreement between the two algorithms. Over
the course of F04, the concentrations of $CH_4$ and $CO_2$ changed, both within the plume (downwind of the plant) as well as in
background air masses (upwind of the plant). Based on available information, it is unknown whether changes in facility
emissions contributed to the observed changes in the plume non-stationarity. It was noted that during sampling, facility
operators instructed researchers that future flights could only sample the boundary of the facility. Operating conditions were
not provided by industry. $NO_2$ and $SO_2$ emissions are often used as tracer data for upscaling $CH_4$ emissions (Baray et al. 2018;
Li et al. 2017; Liggio et al. 2019). Continuous emissions monitoring system (CEMs) facility stack emissions during F04 were
consistent with typical operations and showed $NO_2$ emissions spiked at the beginning of the day, before the aircraft
measurements, and then continuously decreased. Furthermore, flaring data, including volume of gas flared and $SO_2$ emissions,
did not suggest unusual operations at the plant on the day of the flight. The disagreement between the two algorithm estimates
for F04 arise from the non-stationary emission plume which affected the box-flight mass-balance algorithms.

To test the effect of assumptions associated with plume shape below the lowest flight lap various surface extrapolations were
applied to lowest bin of the SciAv divergence profiles for all five flights. The set of all results, based on the differing surface
extrapolations was compiled (S.I., Section 1.1 and 1.3) and estimates are plotted together in Figure 7. Estimates that clustered
together for both methods indicate a good agreement with little difference between the varying surface extrapolation estimates.
F04 has large disagreement between algorithms for both $CH_4$ and $CO_2$ (Figure 7). The mean emission estimate and standard
deviation of each method's various surface extrapolations were calculated (S.I., Section 1.4). The larger the spread in estimates,
the more sensitive the flight was to the choice in extrapolation. Systematic bias is not evident in the differences between
algorithms as emission estimates intersect and no one method produces consistently larger, or smaller estimates for all flights.
To remove the effect of the choice in surface extrapolation, estimates were produced by background mixing ratios below the
lowest flight path. These estimates were compared and the SciAv and TERRA methods were still found to agree (S.I., Section
1.4).





**Figure 7: CH$_4$ and CO$_2$ estimates, based on various surface extrapolation fits, are plotted in purple for the SciAv estimate and green for TERRA. Error bars are drawn onto each algorithm's standard estimate.**

To assess the sensitivity of emission estimates using different surface extrapolations, the differences between each algorithm were calculated for the same four surface extrapolations (S.I., Section 1.4). For most flights, the choice in surface extrapolation had only a small effect on the difference between the estimates ($\leq 3\%$). The choice in surface extrapolation is a source of large variation between the algorithms for F01, the one flight with large emissions at the lowest flight path. The average of the estimates using the same four surface extrapolations was also computed. There is no evidence that agreement changes when removing the effect of surface extrapolation, and there is consistent agreement between the algorithm estimates (S.I., Section 1.4).





For each flight, algorithm estimates for the whole set of varying surface extrapolations were resampled using the bootstrap method (described below) to estimate the range in the difference between estimates based on the various surface extrapolations for each flight. The difference between the two algorithms, based on the various surface extrapolation estimates, was computed and contrasted with the standard error of each estimate. In Figure 8, the distribution of randomly sampled mean difference was calculated using the bootstrap method and plotted along with the propagated error range for each flight and gas ($CH_4$ and $CO_2$).

A value of zero implies that there is no difference between the methods. Aside from F04, the distributions all either include zero, or the error of the standard estimates include zero, indicating that there is good agreement between the algorithms in most cases.



**Figure 8: Distributions of the mean difference between all fits of CH₄ and CO₂ for the SciAv and TERRA algorithms are shown as a light blue histogram. The mean difference between the standard estimates is plotted as a teal dashed lined, and the range in the difference between standard estimates is shown as a light teal box. A grey dot dashed line is drawn at zero as a reference point for the location of exact agreement between the algorithms.**

**3.2 AVIRIS-NG Aircraft Emissions Estimates**

$CH_4$ enhancements were imaged at the Syncrude Plant site within the perimeter of the Scientific Aviation box-flight path of F04 at 21:17:24 UTC on 08/11/2017 (Figure 9) three days prior to the SciAv flight. There appeared to be 2 separate source plumes on that day, both of which were well inside the mass balance transects flown by SciAv during F04. NASA-JPL provided data, analysis, and plume imagery using the methods described in Duren et al. (2019) over the F04 site to help provide additional context for the aircraft measurements (Table 4). The average instantaneous $CH_4$ emission rate of estimates derived





using three different wind speed and direction datasets was 1,665 (kg h$^{-1}$) with an average uncertainty of 707 (kg h$^{-1}$). This average emission rate likely reflects day-to-day emissions variability, as it was approximately 5 times larger than emissions measured using the SciAv method three days later (349 kg h$^{-1}$). However, the AVIRIS-NG derived emission rate was significantly less than the SciAv Syncrude perimeter estimate (F01, 3,840 kg h$^{-1}$ and F05, 3470 kg h$^{-1}$).

**Table 4. AVIRIS-NG data captured on August 14, 2017, three days prior to the F04 flight is estimated using three sources of wind data.**

| Estimate Wind Source | Average Wind Speed (m s$^{-1}$) | Wind Speed Uncertainty (m s$^{-1}$) | CH$_4$ Estimate (kg h$^{-1}$) | CH$_4$ Estimate Uncertainty (kg h$^{-1}$) |
|---|---|---|---|---|
| Met 3062696 | 3.52 | 0.42 | 1767 | 744 (42%) |
| Met 3062697 | 3.80 | 0.64 | 1907 | 834 (44%) |
| MERRA2 reanalysis | 2.62 | 0.235 | 1320 | 543 (41%) |
| Average | | | 1665 | 707 (41%) |

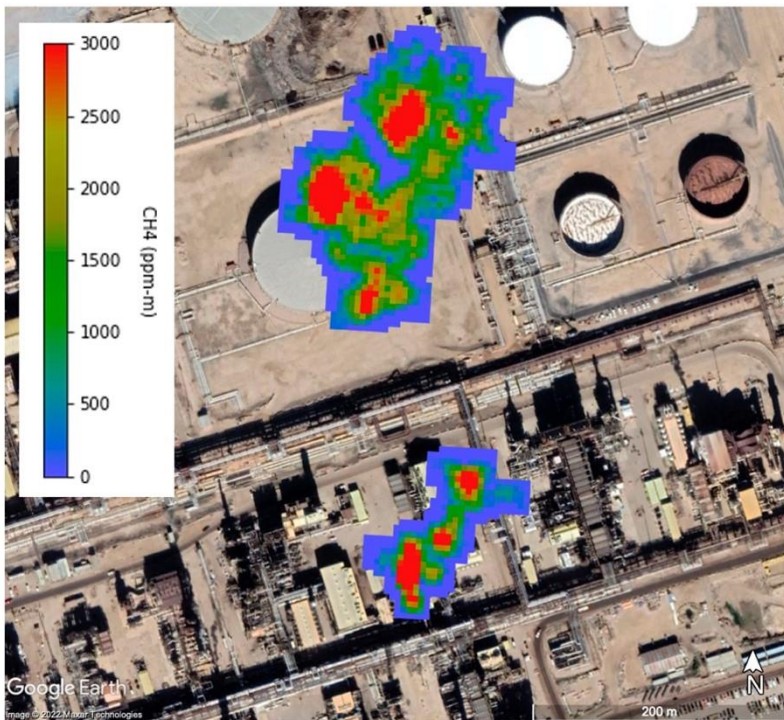

**Figure 9: AVIRIS-NG observed CH$_4$ enhancements were imaged at the Syncrude Plant site on 21:17:24 on 08/11/2017. There appears to be two distinct sources located in close proximity. Satellite imagery © 2020 Maxar Technologies, Google Earth.**



## 4 Discussion

### 4.1 Box-Flight Emissions Estimate Comparisons

In general, when fundamental assumptions were met the SciAv and TERRA algorithms produced similar results. For the average flight scenario, the algorithm estimates derived using various surface extrapolations tended to agree regardless of how the surface extrapolation was fit. This consistency between estimates provides larger certainty in the estimates and in the top-down regional budgets that are inferred from them, and also implies that emissions estimates between studies using different algorithms can be compared.

The SciAv and TERRA estimates were also compared when no surface extrapolation was applied. These estimates also agreed which suggests that the first steps in the core mass-balance algorithm produce similar outputs. Results from applying multiple surface extrapolations indicates that a potential difference between the methods may occur in the second algorithm step, due to the different methods of extrapolating to the surface, when an emission plume is increasing towards the surface.

Plots of the SciAv lap divergence estimates tend to follow three profile types. Examples of the three profile types for measured $CH_4$ lap enhancements (with error) shown in Figure 10 are: 1) an emission plume with constant enhancements persisting at the lowest flight track (Type I); 2) an elevated emission plume where enhancements approach zero at lower altitudes (Type II), and; 3) a plume that has enhancements increasing towards the surface (Type III). All the profiles shown in Figure 10 fully capture the top of each emission plume. Of the five sample flights compared, three had clear profile shapes. The SciAv method of extrapolating as a constant is the most appropriate choice unless a Type III pattern of increasing emissions at the lowest flight path is evident.

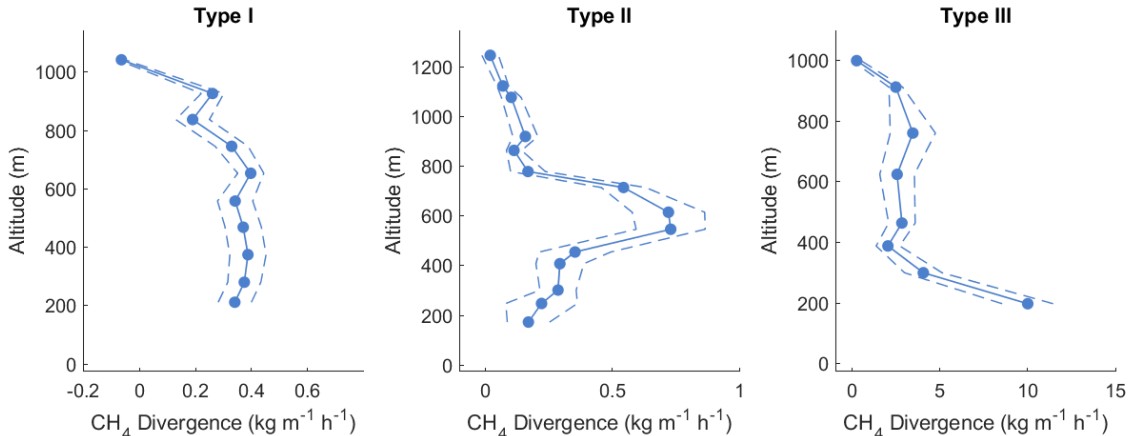

**Figure 10: Sample data representing the three common types of divergence profiles for the SciAv method. Blue dots represent the divergence of each lap with the associated error drawn as blue dashes.**



F01 was the only flight with a definitive SciAv profile type III and highlights the difference in the approaches to surface extrapolation when emissions are increasing at the lowest flight track. An "increasing to surface" fit for profile III shapes would likely improve estimate accuracy for the SciAv algorithm. SciAv fits a constant surface extrapolation regardless of the behaviour of emissions at the lowest flight path, whereas TERRA choses the surface extrapolation from various fits by assessing the plume. When a flight has large emissions at the bottom of the plume, the SciAv and TERRA methods agree more when some form of increasing to surface extrapolation is applied to SciAv compared to use of the standard, constant extrapolation. This indicates that the second step of each algorithm, the choice of an appropriate surface extrapolation, is likely a significant factor contributing to any differences between the methods for type III profiles (see Figure 10).

For the five flights compared, the error from TERRA is lower than that from the SciAv by an average of 8%. The smaller error for the TERRA estimates may be due to how each algorithm quantifies error. TERRA calculates seven specific error terms to address the error of these assumptions. Increases in the error of one assumption does not directly increase the error of others (see S.I., Section 1.2). Whereas, SciAv uses two main broad terms, a temporal error term to capture the extent of stationarity, and a divergence error term to estimate capture of the plume. The divergence error of SciAv can become very large when only a few flight laps are flown. The uncertainty in the surface extrapolation from the SciAv algorithm may be reduced if it is decoupled from the current divergence error term and calculated following TERRA methods (i.e., using the maximum percentage change between probable fits).

The mean of the estimates derived using the six different surface extrapolations were calculated for each integration method's set of results, and differences were tested using a pairwise t-test and Wilcoxon signed-rank test. There was no evidence of a difference in mean estimates for each flight between the two integration methods evaluated (binning vs. trapezoidal; S.I., Section 1.5). This indicates that for this adaption of the SciAv algorithm, choosing a different surface extrapolation is more important than the type of integration method. To further assess the effects of applying different surface extrapolation options, surface measurements at the time of sampling, and more information about the behaviour of a plume, is the most likely path towards further reducing the uncertainty for both algorithms.

## 4.2 Box-Flight Algorithm Assumptions Investigated

During the initial screening of F04 by members of Scientific Aviation and AEP, the non-stationary plume was not identified as focus was placed on assessing meteorological conditions and plume capture. Meteorological conditions are often the most likely source of non-stationarity and as such both SciAv and TERRA methods apply a set of criteria to screen samples. TERRA also assesses conditions using explicit error terms (S.I., Section 1.2). Prior to the ad-hoc analysis of splitting the flight apart, the only measurement that might have indicated atmospheric instability was an air density error term calculated in TERRA.



This produced an error estimate (4 - 6%) that was noticeably larger than the other four flights (1% - 0.01%), but not an
unusually high value for the method in general (S.I., Section 1.2).

The change in the emission plume during sampling is apparent in each method when the data were separated into the ascending
and descending flight periods. In SciAv, emission enhancements for the flight laps going up in altitude noticeably differ from
those flying laps going down (Figure 11). In TERRA, ECCC split the flight data into the upward and downward portions which
were noticeably different (S.I., Section 1.6). For samples with many laps ($\geq 20$), separating the SciAv divergence profile into
upward and downward flight components during screening process would help identify non-stationarity of an emission plume.

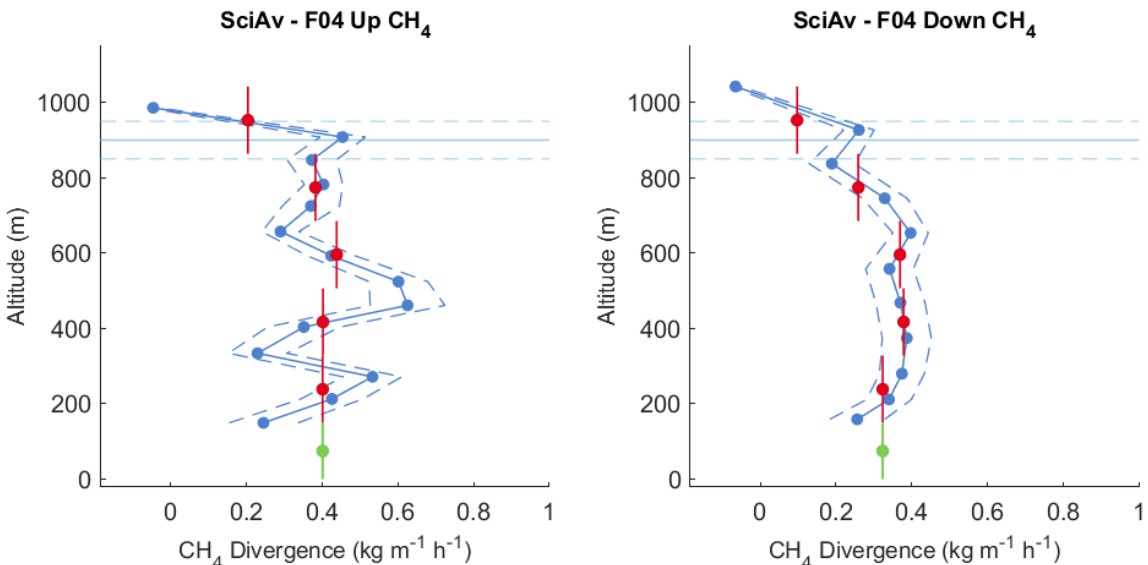

**Figure 11: F04 CH$_4$ divergence profiles for the flight up (left) versus down (right). The profile shapes differ with a much more
variable divergence in the up profile.**

### 4.3 AVIRIS-NG Aircraft Emissions Estimates

Methods based on imaging spectroscopy (e.g., AVIRIS-NG, GHGSat) provide a unique opportunity for emission estimation,
validation of ground-based measurements, and to help develop satellite monitoring techniques, while also providing leak
detection (Cusworth et al., 2019; Frankenberg et al., 2016; Tyner and Johnson, 2021). CH$_4$ emissions from the oil and gas
industry are sporadic, with higher emissions only captured 20-35% of the time when sampling (Duren et al., 2019). While the
SciAv and TERRA methods have lower uncertainties than the AVIRIS-NG method, they require prior knowledge of presumed
sources. Therefore, the mass-balance methods are unlikely to identify unknown sources located outside sampling boundaries.
The persistence of large sporadic emissions, along with their relation to sampling with or without operator notice, should be
studied further. While AVIRIS-NG estimates are dependent on estimates of the wind speed and direction, and repeated



sampling to assess source trends, they avoid the mass-balance requirements of a stationary source and the need to extrapolate emissions to the surface (Duren et al., 2019). This is because AVIRIS-NG effectively samples the entire atmospheric column between the ground and the sensor as a "snapshot".

In our study, the SciAv and TERRA algorithms yielded similar results when proper sampling conditions were met. While non-stationarity occurred for F04, the change in estimates between the upwards and downwards segments does not account for the large difference between the SciAv and AVIRIS-NG measurements of the Syncrude plant. Discrepancies between estimates due to missing, potentially large emissions below the lowest SciAv flight path is highly unlikely as the F04 plume was 'fully captured' with emissions decreasing towards the surface (S.I. Section 1.3). The substantially larger AVIRIS-NG estimate may

be due to industrial operations, such a flaring or venting events. It is possible only one of the two plumes observed by AVIRIS-NG were present when SciAv sampled, or that large day-to-day variability exists in $CH_4$ emissions. However, unlike the SciAv flight, operators were not informed before sampling, and operating conditions were not shared by the facility, so the underlying reason for this difference could not be evaluated. Due to the sporadic tendency of $CH_4$ emissions, and the different sampling date, the AVIRIS-NG result is not directly comparable to the mass-balance algorithms results for F04, but can provide an idea

of the range of potential emissions and specific source locations within the Syncrude Plant. The SciAv and AVIRIS-NG methods have been independently compared and have shown to provide consistently similar emission estimates when employed under similar conditions (Frankenberg et al. 2016; Duren et al., 2019; Thorpe et al. 2020). Given the results of these studies, the AVIRIS-NG data captured on August 14, 2017, may not be anomalously high, but could instead represent independent information on the variability of emissions from the region. Further work comparing these methods under similar

conditions, along with greater transparency in facilities operations, could help confirm these conclusions and support more accurate emission budgets. The large emission estimate from the same site outlines the importance of repeated sampling, and the benefit of using multiple methods to characterize source behaviour, estimate the distribution of emissions from facilities, and estimate regional, national, and global emission budgets.

## 5 Conclusion

We found that emissions estimates were consistent between two top-down mass-balance methods, providing confidence in these methods at a time when emissions reductions are needed. This finding is important because airborne methods are used to validate top-down and bottom-up GHG emissions and develop emissions inventories. The results of this study indicate that, when fundamental assumptions are met, the airborne mass-balance algorithms, SciAv and TERRA, produce similar estimates that agree (3-25%) within algorithm errors (4-34%). The two algorithms disagreed when the fundamental assumption of a

stationary emission plume was not met (F04). Having increased confidence in estimates from the two mass-balance airborne methods provides a more certain foundation for regulatory decisions. Including airborne imaging spectrometer emission estimates in top-down regional budgets can provide additional information about emissions by capturing unknown sources, or



large sporadic emissions. The ideal approach for characterizing and estimating GHG budgets would include repeated measurements, using a combination of airborne methods (in conjunction with new spectroscopic measurements from satellites

for larger, continuous regional estimates), and by using ground-based equipment for small-scale point source quantification (Hardwick and Graven, 2016; National Academies of Sciences, Engineering, 2018; Saunois et al., 2020; Nisbet et al., 2020; Rutherford et al., 2021; Cusworth et al., 2022). Observations that combine and cross-validate multiple monitoring methods at varying scales of sampling will provide the most accurate modeling, improve GHG estimation, and help reconcile the often-reported gap between top-down and bottom-up estimates. Continued advances in developing more accurate inventories will

allow for more effective policy decisions that target the contribution of $CO_2$ and $CH_4$ to climate change.

**Data Availability**

The data files for the five flights can be accessed through the Government of Alberta Portal:

http://ckandata01.canadacentral.cloudapp.azure.com/dataset/aep-noaa-greenhouse-gas-measurement-flights

**Author Contribution**

BE: Conceptualization, Formal analysis, Methodology, Software, Validation, Visualization, Writing – original draft preparation

CA: Supervision, Conceptualization, Investigation, Validation, Methodology, Writing – review & editing

AD: Data Curation, Investigation, Software, Validation, Visualization, Writing – review & editing

MS: Data curation, Formal Analysis, Conceptualization, Writing – review & editing

AT: Formal Analysis, Software, Visualization, Writing – review & editing

GW: Data curation, Investigation, Validation, Writing – review & editing

SC: Resources, Software, Validation, Supervision

JL: Resources, Supervision, Writing – review & editing

SL: Resources, Writing – review & editing

CM: Funding Acquisition, Resources, Writing – review & editing

JG: Supervision, Conceptualization, Validation, Resources, Writing – review & editing

**Competing Interests**

The authors declare that they have no conflict of interest.




**Funding sources**

BE was supported by scholarships from Alberta Innovates and the University of Alberta, and through grants from Natural Sciences and Engineering Research Council (NSERC), and Alberta Environment and Parks. The work was indirectly funded by the Oil Sands Monitoring Program, but does not necessarily reflect an official position of OSM, or of the authors' institutions. Funding was provided to PI C. E. Miller from the Terrestrial Ecology Program via ABoVE for AVIRIS-NG flights.

**Acknowledgments**

The authors wish to thank the pilots of Scientific Aviation for conducting the field work, Dr. Quamrul Huda for contributing to the project design and coordination, and Dr. Bill Donahue for contributing to the project conceptualization and initiation of the Alberta Environment and Parks AEP-NOAA-Scientific Aviation 2017-2018 Alberta Oil Sands Flight Campaign. Thank you to Environment and Climate Change Canada's Mark Gordon for his advice on calculating the TERRA error terms, and Julie Narayan who created the flight path boxes for TERRA. The AVIRIS-NG flights were supported by NASA's Terrestrial Ecology Program's Arctic-Boreal Vulnerability Experiment (ABoVE). Portions of this research was carried out at the Jet Propulsion Laboratory, California Institute of Technology, under a contract with the National Aeronautics and Space Administration (80NM0018D0004).

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
