# Peer review of "Comparing Airborne Algorithms for Greenhouse Gas Flux Measurements over the Alberta Oil Sands"

_Atmospheric Measurement Techniques, 2022_

## Referee Comment (RC1)

Review of Erland et al., Comparing Airborne Algorithms for Greenhouse Gas Flux Measurements over the Alberta Oil Sands.

This is fundamentally a good solid piece of work which is worthy of publication, it is very useful to have data compared in this manner and builds confidence in the methods being applied to convert measurements to fluxes. It is frustrating that activity data isn't available (and I'm sure the authors are more frustrated than me by this) as it would make conclusions and discussion so much stronger. I have a number of minor comments / requests / clarifications which I would like the authors to consider.

L19. Is surveys a better term than samples? Samples gives the impression of very limited data collection.

Abstract general – it would be good to highlight how the lack of on-site activity data can clearly have a profound impact on the depth of conclusions and understanding to such work given the variability of emissions over several days.

L53 (and in other places too). Can the references be split out from the long list so that they match with the parts of the list where they apply?

L69 Feels repetitive from previous paragraph – suggesting rewriting / removing first part.

L80 Replace "Recently, " for "Here," or "In this work,"

L82 un-needed "and"?

L99. Clarification needed here. Coincident sampling stated, but then alludes to periods where sampling conditions are the same… If coincident then should be identical? Maybe needs clarification over what time-period different methods require and therefore over what period conditions need to be stable to give comparable results.

L102. How much is expected to be under reported? A numerical range or approximation would be useful for scale of potential problem.

L106. This is a long sentence and could be reframed within a specific "aims" paragraph? It would be good to mention how this type of work fits with program initiatives such as OGMP v2.0 to encourage operators to properly measure emissions rather than estimating from emission factors / engineering calculations.

L114. The objective feels a bit wooly and non-descript as it is currently written. It would be good to explicitly mention each scheme to be used and reference.

L133 (and other places). Use of the term "components". Component has a very specific meaning in oil and gas emission terms and refers to the smallest level such as valves. These are sub-site level measurements or process level measurements.

L140-142. It would be good to quantify what is sufficient wind, or what constitutes a negligible upwind source etc… in this list.

L157. This is key to the limitations of this study and is a real shame. I feel that the lack of operator buy in should be highlighted to point out how detrimental and hindering it is to scientific conclusions when the operator fails to provide activity data. I feel this should be highlighted to

make the point to policy makers that there should be a mechanism where by this can be requested within reason.

Table 2. Would it be possible to add details to each facility such as total emissions as predicted by inventory / nameplate capacity / age / gas throughput / any other production details?

Paragraph starting L201. As the choice here has some potential for human intervention / error – how much material difference does the choice being made?

L221-222. Description of SA algorithm application insufficient. Can this be referenced or if commercially sensitive please state.

L225. Remove last sentence as not adding anything. Would then move "Figure 3 provides…" to join to the previous paragraph.

L277. As the operators were not informed of this measurement it feels that this measurement is made under a very different sampling protocol, I feel this should be alluded to in the discussion, along with a note on how much pre-warning of measurements they were given for the flight measurements.

Figure 6. With Table 3, is this figure needed in the main text? Could it be in the SI? For table 3, would it be possible to add some information on inventory estimates so that a sense of "potential expected scale of emission" can be seen

L329. Can the reason for no further close flights be expanded on? Is it a safety issue or an operator choice issue?

L376. Can it be clarified that this is different from the emission conditions seen during the flights (i.e. single plume vs dual plume) or whether it was not possible to confirm? If that is the case this is good indication that emission conditions changed between sampling methods and demonstrates nicely the problems of scaling up spot sampling.

L431 (and please check throughout). Use of the term "error" when "uncertainty" is more correct. We do not know the true value so there cannot be known error as such.

L455. The error (see earlier comment!) estimate of 0.01% seems incredibly low. Can this please be checked

L504. Is there anything that can be said about what should be done if fundamental assumptions are broken? Is data collected under non-ideal conditions have any value for determining a flux measurement?

---

## Referee Comment (RC2)

General comments:

The manuscript "Comparing Airborne Algorithms for Greenhouse Gas Flux Measurements over the Alberta Oil Sands" written by Erland et al., provided a technical aspect of estimating the emissions based on airborne measurements. The authors derived the emissions estimates using two algorithms with and without surface extrapolation and found that the two Algorithms agreed well if the ideal conditions over flights met their assumptions. It is very nice to see that the emissions estimates were derived with several surface extrapolation options and the potential reasons why the emissions derived from two algorithms disagree with each other were also discussed. However, I do have a few concerns, which are 1) why are the spectral imaging methods included in the ms and how do they connect to the comparison of two airborne algorithms? It is not clearly explained in the ms. The authors spent a lot of pages through the whole ms, introducing the spectral imaging methods and comparing the emissions derived from them with those derived from mass balance approach. But I do not see the point. That part is more like another independent story, and 2) the specific background information of two algorithms is not sufficiently provided in the introduction. The dis/ad vantages and assumptions of the mass balance approach and remote spectral imaging method are claimed, but when it comes to the two algorithms, the authors simply say the two algorithms are developed and their comparison is not conducted. It is better to list a few studies using the two algorithms and gave a simple summary about the dis/ad-vantages or explanation of them. Overall, the ms is interesting and in a good quality, and therefore, it is suitable to be published in AMT with a minor review.

Specific comments:

Line 54-56: It is very nice to summarize methods for sampling anthropogenic GHG emissions into two major categories and list the literature as support. Could you separate the literature for each category as well? It will be more clear and handier for readers.

Line 69: "the location of emission sources must be known". The expression is too absolute to justify. Most mass-balance flights are used to quantify the emissions from the oil and gas exploring facilities, and to quantify urban emissions from a city. The location of emission sources is very clear for the facilities, but in the case of city emissions, sometimes the location of emission sources is not clear. Before designing the flight, it is expected that the emissions would come from the citywide range (containing multiple source types), but the exact location is unknown, depending which species is of interests. The mass balance flights can detect some missing sources that may not be in the inventories.

Line 75: "Extrapolation to the ground is often the largest error source" referred to two published studies. The authors should mention in which cases the extrapolation is the largest error source since other studies have also pointed out the selection of background and the evolution of the boundary layer were also two major uncertainties. For example, in the paper "Assessment of uncertainties of an aircraft-based mass balance approach" written by Cambaliza et al., (2014), they did sensitivity analysis of estimated fluxes by changing several factors, including two methods for extrapolating the lowest altitude measurements to the surface measurements. They found the extrapolation influenced less than the background and the CBL.

Line 82: delete one of the two verbs? In the sentence "provide two approaches to evaluate calculate mass fluxes"

Line 84-85: "If algorithm comparisons indicate agreement, then emission estimates from multiple campaigns using mass-balance and spectral imaging can be aggregated", but I do not see the logic here. The two algorithms proposed by Gordon et al., (2015) and Conley et al., (2017) both calculate the emissions estimates using mass balance approach, right? I do not understand why spectral imaging is related to this.

Line 117: the authors indicated the second objective. It is interesting but is not in line with the title. Maybe the authors could modify the title to contain the information of the second objective. From my perspective, the first and the second objective do not have necessary connections if the scope of the manuscript is what the title conveys.

Line 125-130: change "divergence" to "flux divergence", just to make it clear to the readers? For the description of "Conceptual Algorithm Steps" of SciAv, you mentioned "divergence" for several times and also in the following texts. The unit of divergence profile shows that the divergence indicates flux divergence. To be honest, I was a bit confused the first time I read it, and after reading the paper by Conley et al., (2017) and the following texts, I understand what it indicated exactly.

Line 220: the subtitle should be 2.2.2

Line 407: "The SciAv and TERRA estimates were also compared when no surface extrapolation was applied". I did not find the results without surface extrapolations. Provide the results of estimates using two algorithms without surface extrapolation.

---

## Author Comment (AC1)

Review of Erland et al., Comparing Airborne Algorithms for Greenhouse Gas Flux Measurements over the Alberta Oil Sands.

This is fundamentally a good solid piece of work which is worthy of publication, it is very useful to have data compared in this manner and builds confidence in the methods being applied to convert measurements to fluxes. It is frustrating that activity data isn't available (and I'm sure the authors are more frustrated than me by this) as it would make conclusions and discussion so much stronger. I have a number of minor comments / requests / clarifications which I would like the authors to consider.

We thank the reviewer for their constructive comments and agree that more detailed activity data would improve the study. Below we provide a line-by-line response, in blue text, to the reviewer's comments.

L19. Is surveys a better term than samples? Samples gives the impression of very limited data collection.

We have opted to keep the term 'samples' here, in order to be consistent with the remainder of the manuscript. Furthermore, although 150 flight surveys ('samples') were completed, only five are considered in this algorithm comparison study. In other words, the amount of data considered is limited relative to the entire survey.

Abstract general – it would be good to highlight how the lack of on-site activity data can clearly have a profound impact on the depth of conclusions and understanding to such work given the variability of emissions over several days.

Agreed – we have added the following text to the abstract to acknowledge this: "In addition, hourly on-site activity data would provide insight on the observed temporal variability in emissions and make a comparison to reported emissions more straightforward."

L53 (and in other places too). Can the references be split out from the long list so that they match with the parts of the list where they apply?

Yes, references have been split out, where appropriate.

L69 Feels repetitive from previous paragraph – suggesting rewriting / removing first part.
This sentence has been removed.

L80 Replace "Recently, " for "Here," or "In this work,"
Changed.

L82 un-needed "and"?
Agreed.

L99. Clarification needed here. Coincident sampling stated, but then alludes to periods where sampling conditions are the same… If coincident then should be identical? Maybe needs clarification over what time-period different methods require and therefore over what period conditions need to be stable to give comparable results.

This is a good point – we have removed the comment about similar conditions and corrected the citation to only include references to the campaign that did coincident sampling.

L102. How much is expected to be under reported? A numerical range or approximation would be useful for scale of potential problem.

Ranges from previous studies have been included as follows: "For example, a recent study aggregated thousands of mobile ground-based emission rate estimates taken without notice to operators from upstream Canadian oil and gas and found that inventories underestimated methane emissions (Atherton et al., 2017; MacKay et al., 2021). Using tower data, methane emission estimates over eight years from oil and gas operations in Western Canada were estimated to be nearly twice those reported in Canada's National Pollution Release Inventory (Chan et al., 2020). Airborne campaigns by ECCC measuring carbon dioxide and methane have also estimated emissions to be 13-123% (Liggio et al., 2019), and 40-56% higher (Baray et al., 2018), respectively than national inventories. In a comparable campaign by Scientific Aviation, industrial upstream oil and gas CH4 emissions estimated in two regions in Alberta were 5 and 17 times higher than values reported to the Alberta Energy Regulator (Johnson et al. 2017)."

L106. This is a long sentence and could be reframed within a specific "aims" paragraph? It would be good to mention how this type of work fits with program initiatives such as OGMP v2.0 to encourage operators to properly measure emissions rather than estimating from emission factors / engineering calculations.

Agreed – this run-on sentence has been segmented and made more concise. The text now reads: "As part of the Joint Oil Sands Monitoring (JOSM) mandate to advance the understanding of Alberta's emissions, a collaborative study was initiated in 2017 by Alberta Environment and Parks (AEP) and the U.S. National Oceanic and Atmospheric Administration (NOAA), contracted to Scientific Aviation. The goal was to use airborne measurements to quantify facility- and activity-specific GHG emissions from mineable and in situ oil sands developments in northern and east-central Alberta."

L114. The objective feels a bit wooly and non-descript as it is currently written. It would be good to explicitly mention each scheme to be used and reference.

We have rephrased this objective to make it more specific: "Our main research objective was to test if emissions estimates from the TERRA and SciAv algorithms agreed within uncertainty,…"

L133 (and other places). Use of the term "components". Component has a very specific meaning in oil and gas emission terms and refers to the smallest level such as valves. These are sub-site level measurements or process level measurements.

Thank you for pointing this out. In order to avoid confusion, we have replaced the term "components" with more appropriate terms (e.g., source area) throughout the manuscript.

L140-142. It would be good to quantify what is sufficient wind, or what constitutes a negligible upwind source etc… in this list.

It is challenging to choose quantitative metrics to classify winds as 'sufficient' or upwind sources as 'negligible', since the relative impact of low wind speeds and upwind sources depends on magnitude of emissions inside the box. Nonetheless, we have provided additional detail on the screening process: "It was challenging to determine a quantitative threshold for adequate wind conditions or negligible upwind sources, since the relative impact on the calculated emission rate depends on magnitude of emissions. Nonetheless, any flight segments with average wind speeds below 5 m s$^{-1}$ were flagged (Gordon et al., 2015), as were flight segments with upwind

mixing ratios above background and assessed further using professional judgement. It is important to note that light/variable winds will increase the uncertainty of SciAv algorithm by increasing the variability between laps."

L157. This is key to the limitations of this study and is a real shame. I feel that the lack of operator buy in should be highlighted to point out how detrimental and hindering it is to scientific conclusions when the operator fails to provide activity data. I feel this should be highlighted to make the point to policy makers that there should be a mechanism where by this can be requested within reason. Would it be possible to add details to each facility such as total emissions as predicted by inventory / nameplate capacity / age / gas throughput / any other production details?

We agree with the reviewer, and recognize the challenges in reconciling annual reported emissions to these hourly emission estimates without detailed, high-time resolution (hourly), activity data. Although this would make the conclusions much stronger, it is out-of-scope for this particular paper which focuses on comparing two similar measurement techniques. Nonetheless, a comparison of top-down hourly emissions from this flight survey to report bottom-up emissions may be the subject of a subsequent paper. The reported $CO_2$ and $CH_4$ annual emissions are available through Canada's Greenhouse Gas Reporting Program data mart (https://open.canada.ca/data/en/dataset/a8ba14b7-7f23-462a-bdbb-83b0ef629823). Monthly production details are available through the Alberta Energy Regulator (https://www.aer.ca/providing-information/data-and-reports/statistical-reports/st39). However, since the objective of this manuscript is method comparison, we have opted to not include details such as reported emissions or production rates in the manuscript.

Paragraph starting L201. As the choice here has some potential for human intervention / error – how much material difference does the choice being made?

The potential for human error can be quite large if a large human intervention occurred and an inappropriate surface extrapolation is chosen, say a background extrapolation when in fact the plume is increasing to surface. This error when comparing methods has been addressed using the bootstrap analysis to compare the difference between all possible surface extrapolation fits (see Figure 8). The text near this section has been update to guide the reader: " All extrapolation outcomes were produced to calculate the surface extrapolation error, which accounts for potential differences when choosing the best surface extrapolation (Gordon et al., 2015), and to compare with the range of possible outcomes from the SciAv method by running a bootstrap analysis."

L221-222. Description of SA algorithm application insufficient. Can this be referenced or if commercially sensitive please state.

The algorithm itself is proprietary; however, we have added the following text to clarify: "Although the algorithm itself is proprietary, the concepts and formulae underpinning the algorithm are described in detail in Conley et al. (2017)". A general description of the algorithm is also given in Table 2.

L225. Remove last sentence as not adding anything. Would then move "Figure 3 provides…" to join to the previous paragraph.
This has been changed.

L277. As the operators were not informed of this measurement it feels that this measurement is made under a very different sampling protocol, I feel this should be alluded to in the discussion, along with a note on how much pre-warning of measurements they were given for the flight measurements.

Agreed - this may be an important difference between the five flights discussed earlier and the AVIRIS-NG data. However, there is no indication that production levels and/or emissions were altered as a result of operators being informed of sampling. Nonetheless, the potential that prior knowledge of sampling could affect emission rates cannot be ruled out, hence this difference is noted.

Figure 6. With Table 3, is this figure needed in the main text? Could it be in the SI? For table 3, would it be possible to add some information on inventory estimates so that a sense of "potential expected scale of emission" can be seen.

Figure 6 has been retained in the main body of the manuscript as we feel it presents the main results of this manuscript and we feel it is useful to illustrate the agreement between the estimates in a way that Figure 7 does not provide. As with the previous comment we agree that inventory estimates would help to give a larger sense of the picture, however this was unfortunately out-of-scope for this paper.

L329. Can the reason for no further close flights be expanded on? Is it a safety issue or an operator choice issue?

Correct – it was a safety issue that was flagged by the operator.

L376. Can it be clarified that this is different from the emission conditions seen during the flights (i.e. single plume vs dual plume) or whether it was not possible to confirm? If that is the case this is good indication that emission conditions changed between sampling methods and demonstrates nicely the problems of scaling up spot sampling.

The TERRA emission screens and Scientific Aviation data were assessed and look to indicate that two plumes were captured during the F04 flight. This would indicate that intensity of the emission plumes changed between methods. The following was added for clarity: "The F04 flight also appeared to capture these two plumes (see Figure S 7)."

L431 (and please check throughout). Use of the term "error" when "uncertainty" is more correct. We do not know the true value so there cannot be known error as such.
This has been changed where appropriate – e.g. the term "error" was kept when referencing "error terms".

L455. The error (see earlier comment!) estimate of 0.01% seems incredibly low. Can this please be checked.

This was checked against another source of data collected near the flight and was not found to be significantly different. The estimate is not unusual for the method as it tens to be low Previous studies using the TERRA method (see Gordon et al. 2015, Baray et al. 2018) and have reported numerous air density uncertainties as 0 (rounded to the nearest integer). This term tends to be a small component of the uncertainty (see Fathi et al. 2021).

L504. Is there anything that can be said about what should be done if fundamental assumptions are broken? Is data collected under non-ideal conditions have any value for determining a flux measurement?

This work highlights the large uncertainty when the algorithms are applied under non-ideal conditions (i.e., when fundamental assumptions are broken), which may not be captured by the uncertainty terms in the algorithms (i.e., the emission rates did not agree within uncertainty). Therefore, there is limited value in using data from non-ideal conditions for evaluating fluxes. This issue can be mitigated by careful flight planning and only sampling when conditions are ideal.

**References:**

Baray, S., Darlington, A., Gordon, M., Hayden, K. L., Leithead, A., Li, S.-M., Liu, P. S. K., Mittermeier, R. L., Moussa, S. G., O'Brien, J., Staebler, R., Wolde, M., Worthy, D., and McLaren, R.: Quantification of methane sources in the Athabasca Oil Sands Region of Alberta by aircraft mass balance, Atmos. Chem. Phys., 18, 7361–7378, https://doi.org/10.5194/acp-18-7361-2018, 2018.

Conley, S., Faloona, I., Mehrotra, S., Suard, M., Lenschow, D. H., Sweeney, C., Herndon, S., Schwietzke, S., Pétron, G., Pifer, J., Kort, E. A., and Schnell, R.: Application of Gauss's theorem to quantify localized surface emissions from airborne measurements of wind and trace gases, Atmos. Meas. Tech., 10, 3345–3358, https://doi.org/10.5194/amt-10-3345-2017, 2017.

Gordon, M., Li, S. M., Staebler, R., Darlington, A., Hayden, K., O'Brien, J., and Wolde, M.: Determining air pollutant emission rates based on mass balance using airborne measurement data over the Alberta oil sands operations, Atmos. Meas. Tech., 8, 3745–3765, https://doi.org/10.5194/amt-8-3745-2015, 2015.

---

## Author Comment (AC2)

General comments:

The manuscript "Comparing Airborne Algorithms for Greenhouse Gas Flux Measurements over the Alberta Oil Sands" written by Erland et al., provided a technical aspect of estimating the emissions based on airborne measurements. The authors derived the emissions estimates using two algorithms with and without surface extrapolation and found that the two Algorithms agreed well if the ideal conditions over flights met their assumptions. It is very nice to see that the emissions estimates were derived with several surface extrapolation options and the potential reasons why the emissions derived from two algorithms disagree with each other were also discussed. However, I do have a few concerns, which are 1) why are the spectral imaging methods included in the ms and how do they connect to the comparison of two airborne algorithms? It is not clearly explained in the ms. The authors spent a lot of pages through the whole ms, introducing the spectral imaging methods and comparing the emissions derived from them with those derived from mass balance approach. But I do not see the point. That part is more like another independent story, and 2) the specific background information of two algorithms is not sufficiently provided in the introduction. The dis/ad vantages and assumptions of the mass balance approach and remote spectral imaging method are claimed, but when it comes to the two algorithms, the authors simply say the two algorithms are developed and their comparison is not conducted. It is better to list a few studies using the two algorithms and gave a simple summary about the dis/ad-vantages or explanation of them. Overall, the ms is interesting and in a good quality, and therefore, it is suitable to be published in AMT with a minor review.

We thank the reviewer for their time on this manuscript. The concerns outline in the general comments have been addressed as follows:

1) Spectral imaging methods were included as useful complementary data to highlight the limitations of using only mass-balance box-flight methods to estimate annual emissions from oil sands facilities. We have updated our secondary objective to make this clearer to the reader and included transitional sentences to further incorporate this part.

2) More background information on the two algorithms has been added to the manuscript. Specifically, a paragraph has been added to the main manuscript before Table 1 to better develop concepts discussed. Two sections have been added to the supplement to expand on the table summarizing the two methods. Equations are included and diagrams have been created to contrast how the two methods derive emissions estimates. The introduction also now includes examples of studies utilizing the methods.

Specific comments:

Line 54-56: It is very nice to summarize methods for sampling anthropogenic GHG emissions into two major categories and list the literature as support. Could you separate the literature for each category as well? It will be more clear and handier for readers.

This has been updated to:

"While some airborne methods utilize eddy covariance measurements (Yuan et al., 2015; Wolfe et al., 2018), methods for sampling anthropogenic GHG emissions from the air tend to fall into two major categories: i) mass-balance methods (O'Shea et al., 2014; Gordon et al., 2015; Conley et al., 2017; France et al., 2021; Foulds et al., 2022) and ii) spectral imaging methods (Duren et al., 2019; Tyner and Johnson, 2021; Krautwurst et al., 2021; Cusworth et al., 2022)."

Line 69: "the location of emission sources must be known". The expression is too absolute to justify. Most mass-balance flights are used to quantify the emissions from the oil and gas exploring facilities, and to quantify urban emissions from a city. The location of emission sources is very clear for the facilities, but in the case of city emissions, sometimes the location of emission sources is not clear. Before designing the flight, it is expected that the emissions would come from the citywide range (containing multiple source types), but the exact location is unknown, depending which species is of interests. The mass balance flights can detect some missing sources that may not be in the inventories.

This line has been removed.

Line 75: "Extrapolation to the ground is often the largest error source" referred to two published studies. The authors should mention in which cases the extrapolation is the largest error source since other studies have also pointed out the selection of background and the evolution of the boundary layer were also two major uncertainties. For example, in the paper "Assessment of uncertainties of an aircraft-based mass balance approach" written by Cambaliza et al., (2014), they did sensitivity analysis of estimated fluxes by changing several factors, including two methods for extrapolating the lowest altitude measurements to the surface measurements. They found the extrapolation influenced less than the background and the CBL.

We agree that there are certainly other large sources of error calculated for the various mass-balance methods. As we are focusing on mass-balance box-flights we have updated this passage to reflect a narrowed focus on that and to outline the fundamental assumptions of the methods more. The update now reads: "For mass-balance box-flights, extrapolation to the ground is often the largest error source, nearing ~30% when the bottom of the plume is not captured (Gordon et al., 2015; Conley et al., 2017). Airborne mass-balance box-flight Mass-balance airborne methods depend on the assumption of a stable boundary layer, and that the emission plume is captured at the top of the box and does not change during sampling (i.e., that conditions are stationary) (Fathi et al., 2021)."

Line 82: delete one of the two verbs? In the sentence "provide two approaches to evaluate calculate mass fluxes"

This has been changed to: "provide two approaches to evaluate mass fluxes"

Line 84-85: "If algorithm comparisons indicate agreement, then emission estimates from multiple campaigns using mass-balance and spectral imaging can be aggregated", but I do not see the logic here. The two algorithms proposed by Gordon et al., (2015) and Conley et al., (2017) both calculate the emissions estimates using mass balance approach, right? I do not understand why spectral imaging is related to this.

We agree and have removed the inclusion of spectral imaging in this paragraph. While we feel multiple methods will improve the estimation of emissions budgets the inclusion of spectral imaging in this paragraph does not fit.

Line 117: the authors indicated the second objective. It is interesting but is not in line with the title. Maybe the authors could modify the title to contain the information of the second objective. From my perspective, the first and the second objective do not have necessary connections if the scope of the manuscript is what the title conveys.

The second objective has been modified to provide a better connection for the reader as to why the information is included in the manuscript and how it relates to the title. It now reads: "Since mass-balance flights are typically flown with the knowledge of and permission from facilities operators, these methods, while they may be accurate, may not necessarily reflect typical operating conditions or GHG emissions. Consequently, a secondary research objective was to examine the potential of utilizing complementary spectral imaging methods, such as AVIRIS-NG, to supplement mass-balance box-flights by providing contextual information to capture the spatial and temporal variability of oil sands GHG emissions." This independent data provides an important check on the representativeness of the mass-balance sampling.

Line 125-130: change "divergence" to "flux divergence", just to make it clear to the readers? For the description of "Conceptual Algorithm Steps" of SciAv, you mentioned "divergence" for several times and also in the following texts. The unit of divergence profile shows that the divergence indicates flux divergence. To be honest, I was a bit confused the first time I read it, and after reading the paper by Conley et al., (2017) and the following texts, I understand what it indicated exactly.

This has been changed.

Line 220: the subtitle should be 2.2.2

This has been changed.

Line 407: "The SciAv and TERRA estimates were also compared when no surface extrapolation was applied". I did not find the results without surface extrapolations. Provide the results of estimates using two algorithms without surface extrapolation.

This has been changed to: "The SciAv and TERRA estimates were also compared when no surface extrapolation was applied (the background surface extrapolation scenario)."

When applying this to the two different methods applying a background extrapolation meant applying no extrapolation below the lowest profile point which means there is only background mixing ratios for Scientific Aviation and applying a background extrapolation to TERRA. This difference is discussed further in Section 1.4 L160 of the supplement where further testing of the "no surface extrapolation" scenario is explored.

Supplement paragraph L160 reads: "To remove the effect of the surface extrapolation the two algorithms were compared when using each method's "background" surface extrapolation fit. For TERRA this meant fitting the chosen background mixing ratio value below the lowest flight lap, and for SciAv calculating zero divergence below the lowest flight lap".

This was removed from the main text to keep the manuscript as concise as possible.